# A Bioinformatics Pipeline to Identify a Subset of SNPs for Genomics-Assisted Potato Breeding

**DOI:** 10.3390/plants10010030

**Published:** 2020-12-24

**Authors:** Catja Selga, Alexander Koc, Aakash Chawade, Rodomiro Ortiz

**Affiliations:** Department of Plant Breeding, Swedish University of Agricultural Sciences (SLU), Box 101, SE-230 53 Alnarp, Sweden; alexander.koc@slu.se (A.K.); aakash.chawade@slu.se (A.C.); rodomiro.ortiz@slu.se (R.O.)

**Keywords:** linkage disequilibrium pruning, genomic selection, genotyping, GWAS, potato breeding

## Abstract

Modern potato breeding methods following a genomic-led approach provide means for shortening breeding cycles and increasing breeding efficiency across selection cycles. Acquiring genetic data for large breeding populations remains expensive. We present a pipeline to reduce the number of single nucleotide polymorphisms (SNPs) to lower the cost of genotyping. First, we reduced the number of individuals to be genotyped with a high-throughput method according to the multi-trait variation as defined by principal component analysis of phenotypic characteristics. Next, we reduced the number of SNPs by pruning for linkage disequilibrium. By adjusting the square of the correlation coefficient between two adjacent loci, we obtained reduced subsets of SNPs. We subsequently tested these SNP subsets by two methods; (1) a genome-wide association study (GWAS) for marker identification, and (2) genomic selection (GS) to predict genomic estimated breeding values. The results indicate that both GWAS and GS can be done without loss of information after SNP reduction. The pipeline allows for creating custom SNP subsets to cover all variation found in any particular breeding population. Low-throughput genotyping will reduce the genotyping cost associated with large populations, thereby making genomic breeding methods applicable to large potato breeding populations by reducing genotyping costs.

## 1. Introduction

Potato is the world’s third most important food crop, with an annual production of more than 300 million ton fresh-weight worldwide [1,2]. Potato is a staple crop for a large portion of the world’s population, and besides being one of the main sources of starch in our diets, it also provides a high amount of protein, minerals, and vitamins [3]. The target traits for breeding are host plant resistance to pathogens and pests, tuber traits, such as weight and number, quality defined by starch content and reducing sugars, and other traits of importance to local producers [4].

Potato has been actively improved through breeding since the 19th century [5]. Despite this, gain in tuber yield from new cultivars has been lagging [6,7]. A challenge in breeding tetraploid table potatoes (*Solanum tuberosum* L.) is the heterozygosity of the plants being used as parents, as it leads to difficulties in predicting the outcome of a cross [8]. The tetraploid potato is crossbred by growing clones over several cycles of phenotypic selection [7]. Each breeding cycle of a new bi-parental population begins with the crossing of two tetraploid parents, producing genetically variable hybrid offspring, wherein each clone has a unique genotype.

In a small to medium-sized breeding program, the number of new potato clones generated each year may vary between 10,000 and 40,000 [9]. Selections done in the first cycle are highly uncertain, as the number of phenotyped individuals per genotype amounts to only one, and the plant is grown from a tuberling that may not be representative for plants used in cultivation [10]. Still, thousands of clones are discarded each year, as the number of clones kept for the second cycle of selection usually are between 10 and 20% of the previous year. It can take up to 12 years from crossing to release of a new cultivar, and it has been estimated that this time could be shortened significantly by implementing molecular breeding methods [11]. Unlike many other crops, potato consumption tends to be based on local production [9]. Hence, breeding new cultivars adapted to a region is of high importance, thereby extending the significance of development for new genetic markers, unique for each breeding population. The early breeding targets are high tuber yield and quality, host-plant resistance—particularly to late blight, caused by the pathogen *Phytophthora infestans*—and early maturity.

Modern plant breeding techniques, such as marker-assisted selection (MAS) are increasingly used in potato breeding. MAS can be used for certain traits, and may lead to a decrease in time spent on phenotyping clones, thus, reducing the time from cross to variety release. Genetic markers are available for a number of phenotypic traits, mostly related to host-plant resistance [12]. Despite the possibility of genetic enhancement, many potato breeding programs have yet to implement MAS in their breeding strategies. The available genetic markers are often limited to association with traits within specific breeding populations with narrow genetic variation [13]. To address this issue, large breeding populations would have to be included in the marker identification process. However, acquiring genotypic data for a large number of potato clones is still very expensive.

With a decline in grant funding, public breeding programs are in demand of more precise and cheaper breeding methods, such as genomic-assisted breeding. A complement, and alternative to traditional MAS using markers linked to a single trait, is genomic selection (GS) by using genomic estimated breeding values (GEBVs) [14,15]. Developing GEBVs for selection seems to be feasible in potato even when there is a significant amount of non-additive genetic variance in elite germplasm [16]. Recently, several reports indicate the applicability of GEBVs as a method of selection for potato breeding. High accuracies of GEBV prediction have been achieved for host plant resistance to late blight and common scab [17], and tuber quality traits, such as starch content or chipping after frying [18,19]. They are based on large training populations to capture the genetic diversity of elite germplasm. Though the costs of high-throughput genotyping have been decreasing over the years, it would still be expensive for a small to medium sized potato breeding program to genotype a breeding population large enough to capture all possible variation for MAS, or a population to be used as a training population for GS.

More genomic resources have become available following the sequencing and assembly of the potato genome in 2011 [20]. One method used is genotyping by sequencing (GBS). GBS is based on reducing complexity of genomes with restriction enzymes and yields high-throughput genomic data. One drawback of GBS is an insufficient number of reads leading to low sequence coverage in the output genotypes [21]. Arrays containing a set of high-quality single nucleotide polymorphisms (SNPs), such as the SolCAP array [22], have also been used in genomic research for potato. Genotype data from SNP arrays usually contains a smaller degree of missing data, and hence requires less data analysis. Furthermore, prediction accuracies for GEBVs do not decrease when reducing the number of genetic markers. A key to success for GS is that the markers are in linkage disequilibrium (LD) with the quantitative trait loci (QTL) for the selected trait [23]. The degree of LD decay in potato is estimated to be relative to other crops indicating that a smaller number of markers is needed to capture all genetic variance [24,25]. Thus, in this study, we propose a pipeline approach to define subsets with reduced number of genetic markers, without losing valuable genetic information (Figure 1). This yields the prospect of lower genotyping costs for large breeding populations for the discovery of new markers for MAS and the application of GS in potato. The results, and implications of the results from the perspective of a small to medium-sized breeding program, are discussed herein.

## 2. Results

### 2.1. Phenotypic Data

Phenotypic data were collected for a breeding population (*n* = 1882) consisting of eight bi-parental crosses from a field experiment in southern Sweden during the spring and summer of 2016. Analysis of variance for the 25 reference clones indicated a higher intra-block error than the inter-block error for all phenotypic traits; hence, no adjustments were required across the field (Appendix A). Host-plant resistance to late blight, measured as area under disease progress curve (AUDCP) affects tuber number per plant, thus, influencing tuber weight at harvest, as shown by the significant coefficient of correlation among them (Pearson’s correlation = −0.467, *p* < 0.001). A subset of the breeding population was selected for high-throughput genotyping. The subset consisted of 11 individuals from each of the eight crosses, which were selected through a principal component analysis (PCA) based on four phenotypic traits, namely: average tuber weight, per plant total tuber weight, per plant total tuber number, and host-plant resistance to late blight, and four parents or grandparents (*n* = 92).

### 2.2. SNP Filtering

Limited population structure was found among the eight crosses based on genotypic data (Figure 2). No population structure among the 92 individuals was revealed for the phenotypic data (Appendix A). From the 12,000 Illumina Infinium V2 Potato SNP Array, 9180 SNPs were called for in the 88 breeding clones and four of their ancestors. To ensure accurate mapping of the SNPs and design of marker assays, a probe basic local alignment search tool (BLAST) against the potato genome was undertaken against the 14 potato pseudomolecules mapped by [26]. SNPs with an identity percentage score below 97% were discarded. Thereafter, 5939 SNPs remained, of which 5122 SNPs were polymorphic with a minor allele frequency (MAF) of 0.05. A number of different thresholds were set by adjusting the squared correlation coefficient of SNP allele frequencies (r^2^) within a set frame of SNPs on each pseudomolecule for LD pruning (Appendix A). A small r^2^ value resulted in fewer SNPs, and increased the average inter-SNP distance per pseudomolecule.

### 2.3. Genome-Wide Association Study

A genome-wide association study (GWAS) was conducted on each of the nine subsets of SNPs obtained from the LD pruning at different thresholds. The population structure was accounted for by including two principal components in the subsequent analysis of the data. Significant QTL were obtained from the GWAS for all phenotypic traits: flowering date (*n* = 2), host plant resistance to late blight (*n* = 1), tuber weight (*n* = 1), and tuber number (*n* = 10) (Figure 3). The total number of significant QTL increased with the number of SNPs in the set (Figure 4). For most of the SNP subsets, the GWAS revealed significant SNPs for all four of the phenotypic traits.

### 2.4. Genomic Prediction

Five priors for controlling shrinkage in genomic prediction were included in this study: Bayesian ridge regression, BayesA, BayesB, BayesC, and Bayesian lasso. For the nine SNP subsets, each prediction model was fitted twice: with and without including family relations as a fixed effect. The highest obtained correlation between GEBVs and observed phenotypic values was 0.24 for host-plant resistance to late blight and 0.20 for tuber number per plant. These values were found in the subsets including 500 and 1500 SNPs respectively (Figure 5). In both cases, the highest obtained correlation between GEBVs and observed phenotype was found when including the family relations as a fixed effect, and using the Bayesian lasso prior. The prediction accuracies of GEBVs did not increase in proportion to the number of markers included in the genotype matrix; however, they seem to be stable for all nine SNP subsets within a range of r^2^ equal to 0.1. No significant differences were found when comparing the r^2^ values between the SNP subsets using the prior with the highest prediction accuracy Bayesian lasso.

## 3. Discussion

We have utilized estimations of LD between adjacent genetic markers as an approach to reduce the marker number for genotyping large potato breeding populations. The proposed stepwise pipeline can be seen in Figure 1. The approach of reducing the number of genetic markers by LD pruning was previously proposed by [27,28]. However, their research undertakings were on genomes of diploid species and very different population structure than in tetraploid potato breeding programs. Estimating LD in tetraploid potato has proven to be a difficult undertaking owing to the outcrossing nature of the species, which leads to a very large range of possible combinations of alleles at each loci [29]. Hence, we decided to take a shortcut by “diploidizing” the genotypic data when pruning for markers in LD. This step included reducing the different types of heterozygote loci from three to one and assuming diploid inheritance. One would assume that this loss of information (the limitation of heterozygotes) would have a negative effect on the accuracy of LD pruning. We think, however, this effect is fairly limited, considering the results from the downstream applications.

In this study, we investigated two genomic-based analyses as validation for a reduced set of SNPs. First, we used a genomic-based association analysis (GWAS). It was recently estimated that 40K SNPs are necessary for successful QTL detection in potatoes [30]. Nonetheless, the high throughput genotyping, which is required to obtain these data would be hard to afford for a large potato breeding population. The SNP subset containing the largest number of SNPs does capture most QTLs. However, the results from the GWAS indicate that it is most cost-effective to use a subset of markers of 1500 SNPs. This SNP subset (*n* = 1500) was able to identify seven QTL, while 12 QTL were found using the biggest subset (*n* = 5000). The number of QTL detected by GWAS stagnated when using a larger set of SNPs, thus indicating that a larger number of SNPs might not be necessary to capture the complexity needed to detect useful markers for MAS in potato. The SNP markers related to QTL found in this study might be of interest for potato breeding. Previous work has described a QTL for host-plant resistance to late blight found on potato pseudomolecule 9 (equivalent to chromosome IX) [31]. The high number of SNP markers related to tuber yield on pseudomolecule 5 could be an indication of a QTL for susceptibility to *P. infestans* on chromosome V [32] as the correlation between these phenotypic traits was high and significant. The matter of proving if the significant SNPs found in this GWAS could be candidate markers for MAS, will have to wait until the remnant of the breeding population has been genotyped with one of the proposed set of SNP markers.

The second genomic-based analysis we used for validation of SNPs reduced by LD pruning was the prediction accuracy of GEBVs. We included two phenotypic traits of high importance to potato breeders—one with a complex genetic background, namely tuber number, and one that appears to be determined by a few, major QTL; i.e., host-plant resistance to late blight [33]. The prediction accuracy for the GEBVs for these two breeding target traits does seem to be affected by the number of SNPs in our population. The prediction accuracies for the traits are low compared to other recent studies, however, this might be due to the very small population used for this study (*n* = 92), and not the set of genetic markers.

Marker pruning by LD seems to be a safe approach to reduce the number of markers used for genotyping potato, thus lowering the costs related to generating genotypic data. Looking at the spread of SNP markers across the potato pseudomolecules, it is obvious that the harshest pruning has taken place close to the centromeres. This decrease of LD further out on the chromosome arms is to be expected [34]. Pruning markers for LD can eliminate redundancy in the data and, thus, shrink the strong influence of SNP clusters [35]. Limiting marker redundancy should be beneficial for both GWAS, with the limitation of false positives, and GS with the shrinkage of individual marker influence.

There are alternative methods for marker reduction, where SNPs are filtered on criteria other than LD correlation. For example, ranking SNPs based on p-values obtained from a GWAS, minor allele frequency or limiting to a single SNP per haplotype block [36]. In this project, we also tried a SNP clumping approach [36] to reduction the number of markers. However, this method failed to produce subsets with defined number of SNPs, as was our aim with this study. The plants in the subset representing the eight biparental breeding populations were selected to cover all phenotypic diversity present therein. The principal component analysis (PCA) that was used to select the plants is an impartial method of selecting representative individuals from a larger population. In the future, to validate this method, the remnant of the breeding population will have to be genotyped using the selected subset of SNP markers. The method of selecting representative individuals, based on variation linked to the phenotypic traits of interest, should ensure the genetic variation to be covered by the SNPs found to be polymorphic in the study. Additionally, including parents and grandparents in the high-throughput genotyping, further opens up the possibility of the inclusion of all possible genetic variation present in the breeding population.

In this study, we show that a genomic-based analysis (GWAS) or breeding approach (GS) for potato can be performed with a marker set reduced by LD pruning. For a small to medium-sized potato breeding program (producing between 10,000 and 40,000 clones in the first breeding generation annually), engaging in genomic-led genetic enhancement may be costly. We have showed that genotyping with a selected subset of the breeding population, and subsequent SNP pruning for LD, is an effect approach to reduce the number of genetic markers while not losing any of the complexity in the genetic information. We think this pipeline would increase the availability of genetic-enhanced breeding techniques for potato genetic enhancement, without limiting the size of the potato breeding population that today appears affordable to genotype.

## 4. Materials and Methods

This experiment included 1882 potato breeding clones (a sample of T1 generation of SLU Potato Breeding Program) representing eight biparental crosses along with some of their parents or grandparents. The experimental layout followed an augmented quadruple lattice design with 25 reference clones and cultivars to ensure uniformity across the field. Such a field design allows accounting for environmental differences across plots. Germplasm was evaluated for four phenotypic traits in the 2016 growing season at field in Mosslunda (55°97′17″ N, 14°11′05″ E) in Southern Sweden. Tuber number and total tuber weight per plant data were recorded at harvest. Flowering time was noted throughout the growth season as number of days until floral emergence. Susceptibility to *P. infestans* was defined by the area under disease progress curve (AUDPC) following the EucaBlight protocol [37]. This host plant resistance recording was done six times throughout the growing season. The analysis of variance, considering between and within family variation, isolated the components of variance from which intra-class correlation (a proxy for heritability) for each trait were estimated in these segregating populations.

### 4.1. Sampling of Breeding Population for Genotyping, DNA Extraction and SNP Genotyping

A total of 88 breeding clones along with four of their ancestors (N = 92) were genotyped using a SNP array. From each of the eight crosses, 11 breeding clones were selected through a principal component analysis (PCA). The PCA was based on tuber number per plant, tuber weight per plant, the estimated individual tuber weight, and AUDCP. Flowering time was excluded due to missing data. The clones were selected based on an even spread across the axis representing the variation in AUDPC (for example, see Appendix A). Extraction of genomic DNA, genotyping, and SNP dosage calling was carried out by Trait Genetics and LGC, respectively. Approximately 15 mg leaf tissue was homogenized by freezing the sample in liquid nitrogen and homogenized mechanically. DNA was extracted using the DNeasy Plant Mini Kit (QIAGEN, Hilden, Germany) following the manufacturer’s instructions. The DNA was quantified with the Quant-it PicoGreen assay (Invitrogen, San Diego, CA) and adjusted to a concentration of 50 ng/mL. The material was genotyped with the Illumina Infinium V2 Potato SNP Array (12,720 SNPs: original SolCAP Infinium 8303 Potato SNP Array with 4500 additional SNPs to increase coverage in candidate genes and R-gene hotspots) [22]. For genotype calling, the Illumina GenomeStudio software (Illumina, San Diego, CA) was used. Tetraploid genotyping was based on theta value thresholds using a custom script from the SolCAP project [38] to call for the five allelic states.

### 4.2. SNP Filtering

Genotypic data from 92 clones of tetraploid potato was obtained from a high-throughput array including 12,000 SNPs. In total, 9180 SNPs with 10 or less missing genotypes were represented in the 92 individuals. These SNPs were mapped to the potato genome sequence of *S. tuberosum* group Phureja DMI-3 516 R44 v. 4.04 with 14 pseudomolecules [26]. SNPs with identity percentage scores below 97% were discarded. The minor allele frequency (MAF) was further calculated using the package GWASpoly (Rosyara et al. 2016) in the R Software [39]. SNPs with a MAF below 0.05 were excluded to ensure marker polymorphism. Linkage disequilibrium (LD) pruning was conducted at nine distinct thresholds, creating nine SNP subsets, using the software PLINK [40] to filter the SNPs. The thresholds were defined by adjusting the square correlation coefficient (r^2^) between independent pairwise markers on each pseudomolecule. The window frame for r^2^ was 50 SNPs, and each new window was set by a move of 10 SNPs. The process of LD pruning also ensures an even spread of SNP markers across the genome. This software is not able to handle five cluster data; thus, the data was “diploidized” reducing the number of heterozygote loci from three (AAAa, AAaa, and Aaaa) to one (Aa). The largest and smallest of the nine SNP subsets contained approximately 5000 SNPs and 500 SNPs, respectively. Population structure was estimated through a principal coordinate analysis (PCoA) on the genotypic data.

### 4.3. Post-Filtering Evaluations

A GWAS was carried out using the R package GWASpoly [41] for each of the four phenotypic traits. The number of significant SNPs were counted for each of the nine SNP subsets created from the LD pruning thresholds. The GWAS was based on four mixed models that considered the population structure and kinship, and was run with the optimum level of compression. The significance of the test was set to 0.05 and Bonferroni correction was applied to adjust for multiple testing.

Models for GS were created for two of the phenotypic traits—per plant tuber number and tuber weight, and AUDPC for host-plant resistance to late blight, for each of the nine SNP subsets using the R package BGLR [42]. For each of the models above, five marker effect priors were evaluated for genomic-estimated breeding values (GEBVs): Bayesian ridge regression, BayesA, BayesB, BayesC, and Bayesian lasso. The four grandparents/parents were excluded from the data to limit the variation in the population. A leave-one-out cross validation approach was used to validate each model combination of marker subset and prior. In addition to this, each of the previous combinations were run twice, once with family relations included as a fixed effect predictor and once without. In each of the model runs, BGLR was run with 10,000 iterations, thinning at 10, and a burn-in setting of 5000 iterations. The prediction accuracy of the models determined as Pearson’s correlation coefficient between observed phenotypes and GEBVs, and the difference of these correlations was tested using Fisher’s r to z transformation with a subsequent z-test [43].

## Figures and Tables

**Figure 1 plants-10-00030-f001:**
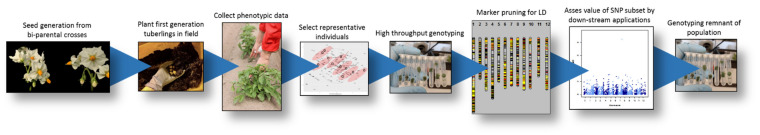
Suggested pipeline for reduced genotyping costs for large breeding populations. Steps include limiting the number of individuals genotyped with costly, high-throughput methods, reduce number of single nucleotide polymorphisms (SNPs) by pruning for markers in linkage disequilibrium, applying genomic-based plant breeding approaches such as genome-wide association study (GWAS) and genomic selection (GS), and genotyping the remnant of the breeding population with a reduced subset of markers.

**Figure 2 plants-10-00030-f002:**
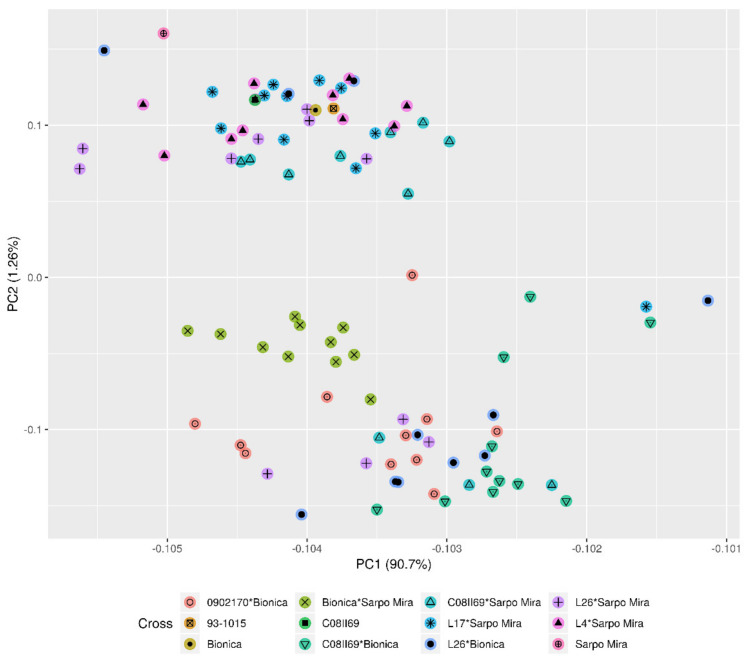
Principal coordinate analysis from genetic kinship among the eight crossing populations and four parents or grandparents (*n* = 92) revealed limited population structure, which was accounted for in the genome-wide association study (GWAS) by adding two principal components.

**Figure 3 plants-10-00030-f003:**
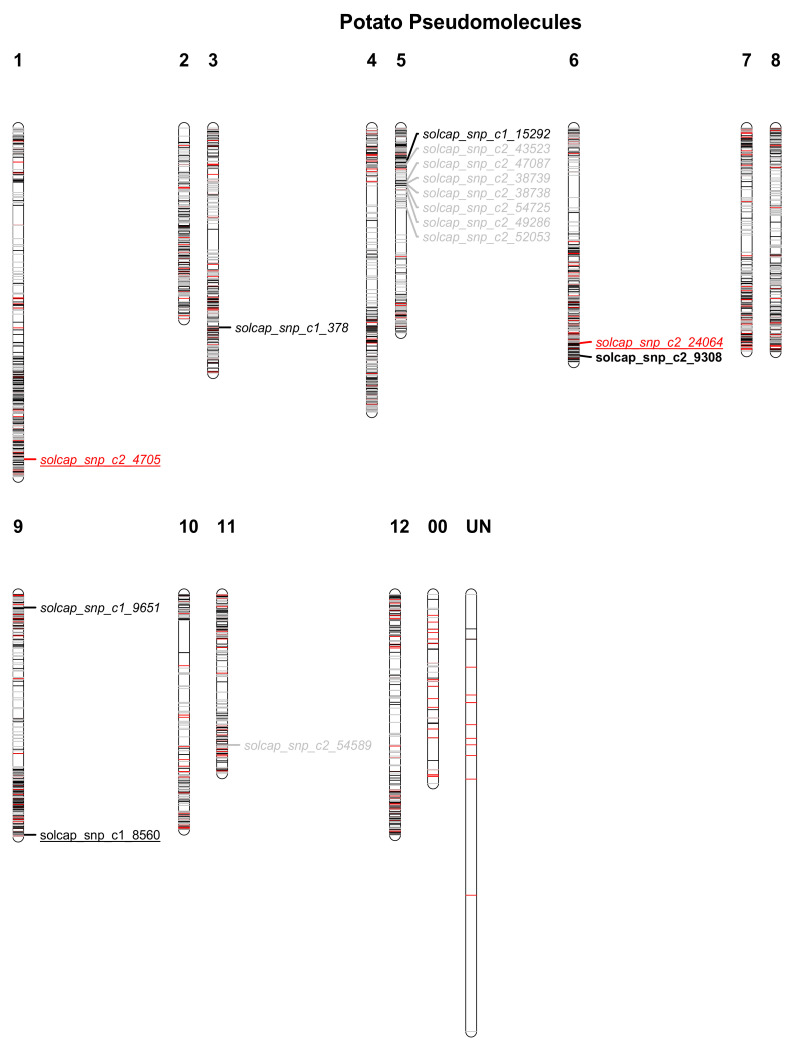
Distribution of SNP markers from three sets of linkage disequilibrium (LD) pruning over the 14 (1–12, 00 and UN) potato pseudomolecules previously mapped. The displayed markers are from three distinct square correlation coefficient (r^2^) LD thresholds for SNP filtering; 5000 SNPs were extracted by LD pruning at r^2^ = 1 (grey), 2000 SNPs at r^2^ = 0.446 (black) and 500 SNPs at r^2^ = 0.084 (red). Significant SNPs from all thresholds are mapped underlined and italic flowering time, italic tuber number, bold tuber weight, and underlined susceptibility to *Phytophthora infestans* causing late blight in potato.

**Figure 4 plants-10-00030-f004:**
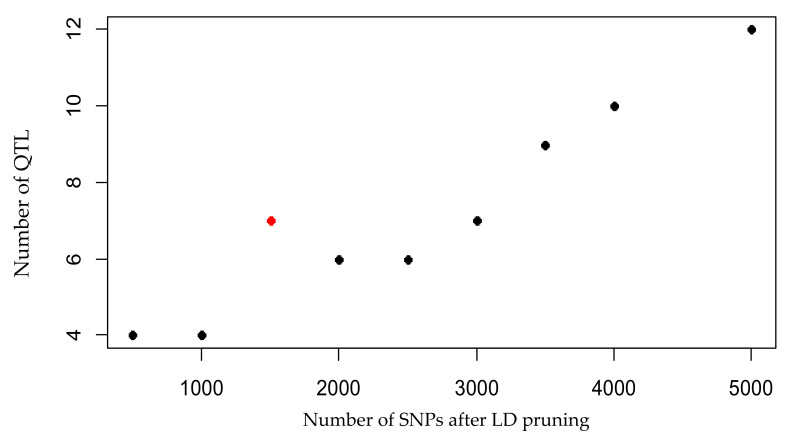
Number of quantitative trait loci (QTL) found after GWAS for each set of SNPs after LD pruning. The red point indicates the maximum number of significant QTLs per SNP found at SNP set with 1500 SNPs (r^2^ = 0.289).

**Figure 5 plants-10-00030-f005:**
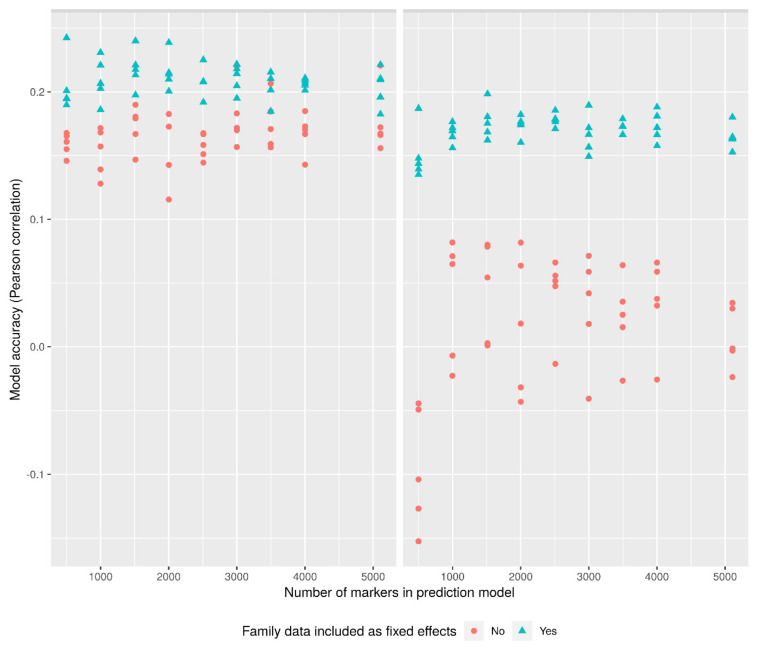
Prediction accuracy (y-axis) of genomic selection models for the nine SNP subsets (x-axis) for two phenotypic traits—host-plants resistance to late blight (left) and per plant tuber number (right). Prediction accuracies were determined as the Pearson’s correlation coefficient between observed phenotypes and genomic estimated breeding values (GEBVs). All five priors for controlling shrinkage in genomic prediction are represented in the nine SNP subsets: Bayesian ridge regression, BayesA, BayesB, BayesC, and Bayesian lasso. Each prediction model was fit twice: with and without including family relations as a fixed effect.

## Data Availability

Not applicable.

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
