# Peer review of "A Bioinformatics Pipeline to Identify a Subset of SNPs for Genomics-Assisted Potato Breeding"

_plants, 2020, doi:10.3390/plants10010030_

Round 1

Reviewer 1 Report

This is a well presented and scientifically sound work. The conclusions, although (as the authors declare) still limited in scope, are interesting and will trigger future works.

The conclusion at line 226 is a bit reckless. Caption to figure 3 is somewhat unclear. Caption to figure 4 should mention what is the red meaning.

The ms. describes a novel pipeline for SNP pruning, based on LD, in polyploid species. The Authors applied their pipeline to potato and successfully proved that their pruning method provides a faster approach to GWAS and GS. This has worked even on a breeding population of reduced size.

This  ms. represents a first attempt of SNP pruning in potato, in other worlds this ms. is not a validation of the pipeline. This was previously done in other species, as reported by the Authors, but on different ploidy and population structure.

This ms. is limited in scope, as it uses a fairly small population. The Authors agree in stating that they need to validate the method using a larger population, possibly including ancestors.

The Authors need to discuss the comparison between their pipeline and alternative ones, such as clumping.

Importantly, the Authors need to clarify what is leading them to the “most likely” conclusion at lines 226 and 227.

Additionally, and on a minor note, they need to better clarify captions to figures 3 and 4 (the first one is somewhat unclear, the second one does not address the red dot).

Author Response

Thank you very much for your comments! We have now adapted our manuscript accordingly. 

  • The Authors need to discuss the comparison between their pipeline and alternative ones, such as clumping.

Authors: A section has been added where alternative methods for marker reductions are discusses, lines 239-243.

  • Importantly, the Authors need to clarify what is leading them to the “most likely” conclusion at lines 226 and 227.

Authors: We agree that this was a bit too drastic a conclusion, the statement on lines 228-230 has been changed.

  • Additionally, and on a minor note, they need to better clarify captions to figures 3 and 4 (the first one is somewhat unclear, the second one does not address the red dot).

Authors: Figure legends has been altered for clarification.

Reviewer 2 Report

They showed that a genomic-based analysis (GWAS) or breeding approach (GS) for potato can be performed with a marker set reduced by LD pruning. It is will useful for a small to medium-sized potato breeding programme engaging in genomic-led genetic enhancement may be too costly. They had already showed that genotyping with a selected subset of the breeding population, and subsequent SNP pruning for LD, is an effect approach to reduce the number of genetic markers while not losing any of the complexity in the genetic information. It is likely this pipeline would increase the availability of genetic-enhanced breeding techniques for potato genetic enhancement, without limiting the size of the potato breeding population.

Author Response

Thank you very much for your feedback. Language and style has been edited. A new paragraph has been added to the discussion (lines:239-243).